# Comparison of quantitative ultrasonography and dual X-ray absorptiometry for bone status assessment in South African children living with HIV

Jackson A. Roberts[1], Yanhan Shen[2], Renate Strehlau[3], Faeezah Patel[3], Louise Kuhn[2,4], Ashraf Coovadia[3], Jonathan J. Kaufman[5,6], Stephanie Shiau[7], Stephen M. Arpadi[2,4,8], Michael T. Yin[9]*

1 Columbia University Vagelos College of Physicians and Surgeons, New York, New York, United States of America, 2 Gertrude H. Sergievsky Center, Vagelos College of Physicians and Surgeons, Columbia University Irving Medical Center, New York, New York, United States of America, 3 Empilweni Services and Research Unit, Rahima Moosa Mother and Child Hospital, Department of Paediatrics and Child Health, Faculty of Health Sciences, University of the Witwatersrand, Johannesburg, South Africa, 4 Department of Epidemiology, Mailman School of Public Health, Columbia University, New York, New York, United States of America, 5 Cyberlogic, Inc., New York, New York, United States of America, 6 Department of Orthopedics, The Mount Sinai Medical Center, New York, New York, United States of America, 7 Department of Biostatistics and Epidemiology, Rutgers School of Public Health, Rutgers University, Piscataway, New Jersey, United States of America, 8 Department of Pediatrics, Vagelos College of Physicians & Surgeons, Columbia University Irving Medical Center, New York, New York, United States of America, 9 Division of Infectious Disease, Department of Medicine, Vagelos College of Physicians & Surgeons, Columbia University Irving Medical Center, New York, New York, United States of America

* mty4@cumc.columbia.edu

**Data Availability Statement:** All relevant data are within the paper and its Supporting information files.

## Abstract

Children living with HIV (CLHIV) have decreased bone mineral content (BMC) and density (BMD), increasing risk for fracture and future osteoporosis. While DXA is the gold-standard for bone assessments, it lacks availability in resource-constrained settings (RCS). Quantitative ultrasound (QUS) offers an alternative owing to its portability, low cost, ease of handling, and lack of ionizing radiation. While QUS has detected reduced bone quality in CLHIV, the relationship between QUS and DXA in this population remains unexplored. At baseline and 12 months, BMC and BMD of the whole body, lumbar spine, and radius were measured by DXA in a longitudinal cohort of CLHIV in Johannesburg, South Africa. Calcaneal speed of sound (SOS) and broadband ultrasound attenuation (BUA) and radius SOS were obtained by QUS, and calcaneal stiffness index (SI) was calculated. Spearman correlations, with and without HIV stratification, were performed between QUS and DXA measurements at each visit and for absolute difference in measurements between visits. At baseline and 12-months, calcaneal BUA and SI displayed strong positive correlations with DXA, with only modest correlations between radial QUS and DXA at baseline. Longitudinal measures of QUS did not correlate with DXA. At both baseline and 12-months, individuals with DXA whole-body BMD *z*-score < -1 displayed significantly lower calcaneal BUA and SI. Cross-sectionally, calcaneal QUS correlates strongly with whole body DXA and may represent a

**Funding:** Funded by the Eunice Kennedy Shriver National Institute of Child Health and Human Development (HD 073977, S.A. and M.T.Y) and National Institute of Allergy and Infectious Diseases (K24 AI155230, M.T.Y). The funders had no role in study design, data collection and analysis, decision to publish, or preparation of the manuscript.

**Competing interests:** The authors have declared that no competing interests exist.

viable diagnostic alternative in RCS. Longitudinally, the two methods do not correlate well, possibly reflecting that each method assesses distinct aspects of bone architecture.

## Introduction

Improved access to antiretroviral therapy (ART) has allowed many individuals with perinatally-acquired HIV (PHIV) to survive into adulthood. Childhood and adolescence represent a critical period for bone growth and turnover during which 85–90 percent of adult bone mass is acquired; therefore, disruptions of bone growth at this stage have the potential to impact long-term bone health [1]. Compared to uninfected individuals, children and adolescents living with HIV (CLHIV) have consistently been found to have decreased bone mineral content (BMC) and density (BMD), which may place them at risk for future osteoporosis and bone fracture [2].

To date, the majority of studies examining bone health in CLHIV have been conducted using dual-energy X-ray absorptiometry (DXA), generally considered the gold-standard. However, availability of DXA is limited in resource-constrained settings (RCS), due to high equipment cost and the requirement for extensively trained personnel for image acquisition and interpretation [3]. Considering that greater than 95 percent of PHIV instances occur in RCS [4], developing simpler low-cost alternative methods for evaluating bone status is an important consideration for research and clinical care in RCS.

A potential alternative is quantitative ultrasonography (QUS). In contrast to DXA, which utilizes two X-ray beams to generate attenuation coefficients attributable to soft tissue and bone, QUS operates by generating a traveling mechanical vibration whose shape, intensity, and speed are altered by the material properties of the bone through which it passes. Both the velocity and attenuation of the signal as it passes through tissue may be obtained non-invasively and without exposure to ionizing radiation to provide an assessment of bone quality.

Compared to DXA, QUS machines are highly portable, low in cost, require short scan times, and require less technical expertise for operation and interpretation [5]. Additionally, broadband ultrasound attenuation (BUA) measured by QUS has previously been shown to predict fracture risk in certain populations, perhaps with even better predictive power than DXA [6, 7]. As such, QUS offers an attractive option for bone status assessment in RCS, but significant challenges to its application remain, including the variability between the wide variety of QUS devices and inconsistent data supporting their use in specific populations and location of measurement sites.

To support its widespread clinical implementation, comparisons of QUS to the gold standard DXA may be informative. Some previous studies have reported inconsistent correlations between QUS and DXA, owing in part to the heterogeneity of QUS devices and study populations examined [7–9]. Few of these correlation studies between QUS and DXA have been performed in RCS. While we have previously shown that calcaneal QUS measures are reduced in CLHIV [10], these findings have not yet been related to DXA measures of bone quality in CLHIV.

To that end, the present work evaluates the correlations over one year between two different methods of QUS and DXA in young South African CLHIV receiving anti-retroviral therapies. Overall, we identified that calcaneal QUS correlates strongly with DXA in cross-section but not longitudinally, while radial QUS demonstrates a weaker correlation with DXA measurements of bone quality.

## Methods

### Study population

Subjects included in this analysis were a subset of participants from the Childhood HAART Alterations in Normal Growth, Genes, and aGing Evaluation Study (CHANGES) cohort, a longitudinal study conducted at Rahima Moosa Mother and Child Hospital and Chris Hani Baragwanath Hospital in Johannesburg, South Africa. Details of recruitment have been previously published [11, 12]. Enrollment occurred between March 2014 and May 2016. CLHIV (n = 218) were participants in a prior noninferiority randomized clinical trial (RCT) comparing viral suppression between initially viral-suppressed children who switched to EFV-based therapy and those remaining on LPV/r-based therapy [13]. Controls without HIV (n = 205) were recruited from siblings and household members of CLHIV [11]. Participants included in this study had data collected at baseline and 12-months. The study was approved by the Institutional Review Boards of Columbia University Irving Medical Center (CUIMC; New York, NY, USA) and the University of Witwatersrand (Johannesburg, South Africa). Signed informed consent was obtained from each child's parent or guardian, and children provided assent if at least 7 years old and deemed able to understand.

### Measurements and procedures

Digital scale and wall-mounted stadiometer were used to obtain anthropometric measurements [11]. Weight-for-age $z$ score (WAZ), underweight (WAZ $\leq$ -2), overweight (WAZ > 2) for children aged 5–10 years old, height-for-age $z$ score (HAZ), stunted (HAZ $\leq$ -2) for children aged 5–19 years old and BMI-for-age z score (BAZ) were determined using WHO norms [14]. Trained physicians assessed stage of pubertal development by using the highest score of breast or public hair development for female individuals and pubic hair for male individuals according to Tanner Staging method [11]. For CLHIV, plasma HIV-RNA levels (copies/ml) were determined with the Abbott RealTime HIV Assay (Abbot Park, Illinois, USA) and CD4 + counts (cells/µl and %) were measured by TruCount Method (BD Biosciences, Germany). Comparisons between ART regimens were performed only with those on a consistent regiment for the full study period.

### Dual-energy X-ray absorptiometry (DXA)

Whole body (WB) and lumbar spine (LS) DXA scans were performed by licensed radiographers using a single Hologic Discover Wi bone densitometer at the Department of Radiology of Rahima Moosa Mother and Child Hospital. Scans were analyzed by a single technician, blinded to HIV status, using Apex software version 3.4 (Hologic Inc., Bedford, MA, USA) at the CUIMC Body Composition Unit (New York, NY, USA). Output measures included bone area in cm$^2$, bone mineral content (BMC) in grams, and bone mineral density (BMD) in grams/cm$^2$ [15]. BMD is a two-dimensional parameter and has potentially important limitations in growing children, as it may be strongly influenced by bone size [16]. To address this, BMD and BMC $z$-scores adjusted for age, sex, race and HAZ were calculated using reference norms from the US Bone Mineral Density in Childhood Study[17], as no South African references were available.

### Calcaneal quantitative ultrasonography (QUS)

Calcaneal QUS measures were obtained at each visit using a Lunar Achilles Insight device (GE Healthcare, Madison, WI, USA) by study physicians using the standard operating procedures indicated by the manufacturer's operating manual [18]. The built-in quality assurance test was

performed prior to each use to ensure proper unit temperature, water level, and coupling. All measurements were obtained using manufacturer-provided foot shims to improve positioning. The device's moveable region of interest (ROI) and the real-time QUS image were used to determine the size and location of the ROI, as well as the proper positioning of the foot [18]. Calcaneal measurements included broadband attenuated ultrasound (BUA) in decibels/megahertz (dB/MHz) and speed of sound (SOS) in meters/second (m/s). Calcaneal stiffness index (SI) was output from these measures using Njeh *et al.*'s adapted equation: SI = (0.67 * BUA + 0.28 * SOS)– 320 [19].

We excluded participants with SOS values $\geq$ 1625 m/s at either baseline or 12-months, as we have previously noted that such values represent spurious readings due to small foot size [20].

## Radial quantitative ultrasonography

QUS measures were obtained along the distal 1/3 radius using the Sunlight Mini-Omnisense Sonometer (BeamMed, Petah Tikva, Israel). The primary output of this device is a transaxial measurement of SOS in m/s. To take measurements the probe of the device was positioned at the midpoint between the elbow and tip of the middle finger, parallel to the axis of bone. Once aligned, the probe was moved perpendicularly to the bone axis both medially and laterally, as described previously [21].

## Statistical analyses

Chi-squared tests were used to compare categorical variables between groups. Student *t* tests and Wilcoxon rank-sum tests were conducted for normally and non-normally distributed measurements, respectively. Absolute change was calculated as the difference between measurements at 12-months and baseline. For missing data at the 12-month visit, data was imputed from a 6-month study visit using last observation carried forward imputation as a sensitivity analysis. We conducted an additional sensitivity analysis for individuals with missing data in which we performed multiple imputation with the *mice* package in R [22] with the assumption that data were missing at random.

Spearman correlations were performed to determine the correlation between QUS and DXA measurements. Power calculations were performed for both calcaneal and radial QUS correlations utilizing the *pwr* package in R [23]. For these calculations, desired power was set at 80 percent, alpha level was 0.05 and sample sizes were input at those present in this study. Correlations were performed at each visit separately, as well as for the absolute change in QUS and DXA measurements between visits. In the main analysis, correlations were pooled across HIV status. Correlation analyses stratified by HIV infection status were conducted. We additionally performed within-subject Spearman correlations for calcaneal BUA and whole body DXA BMD across baseline and follow-up visits, stratified by HIV infection status.

We also assessed if individuals with abnormal BMD as determined by DXA demonstrated differences in Calcaneal QUS measurements. To achieve this, *z*-scores were calculated for participants at each visit as previously described [24], adjusting for age, sex, race, height-for-age *z*-score and whole-body weight less head *z*-score. We established *z*-score groups by 3 methods: those with $z \leq$ -2.0, consistent with international consensus guideline definitions for childhood and adolescent osteoporosis [25], as well as those with $z \leq$ -1.0 or those with a decline in *z*-score between two visits. We then compared QUS parameters between these groups using Student's t-tests.

## Results

### Demographic characteristics

Of the 423 individuals with DXA scans at baseline and 12-months in this study, 279 had calcaneal scans at baseline and 12-months. There were few significant differences between the 279 individuals with QUS scans and the 144 individuals without QUS, though a lower proportion of those without scans were CLHIV (S1 Table in S1 File). Of these 279, 109 had an abnormally elevated SOS value ($> 1625$ m/s) at either baseline or 12-months and were excluded. Differences between those included in analyses and those excluded are presented in S2 Table in S1 File. This resulted in a final sample of 80 CLHIV and 90 controls for calcaneal QUS analyses. At both baseline and 12-month follow-up, CLHIV had lower weight and height and were more often categorized as stunted compared to uninfected controls (Table 1). Both groups were similar in terms of age, sex, categorization as underweight, BMI-for-age $z$-score, and Tanner stage.

Of the original 423 participants, 205 had radial scans at both baseline and the 12-month follow-up. Compared to those with scans, those missing radial QUS scans had lower average height, higher BMI-for-age z-score, and lower CD4 cell count (S3 Table in S1 File). This resulted in a final sample of 101 CLHIV and 104 controls for radial QUS analyses. At baseline and 12-month follow-up, CLHIV had lower weight and height, were more often categorized as underweight, and more often categorized as stunted (Table 2). Both groups were similar in terms of sex, BMI-for-age $z$-score, and Tanner stage at both baseline and follow-up, while uninfected controls were significantly older at follow-up.

We additionally conducted sensitivity analyses by imputing missing data for Tanner staging using the last observation carried forward method, in which we utilized Tanner stage from a 6-month follow up for those missing data at 12-months. Results remained similar after imputation, as 15 of 16 individuals with missing data were Tanner Stage 1 at the 6-month visit. Similarly, for those missing viral load data at 12-months, we imputed data from the 6-month visit, which was available for 14 of 16 participants with missing data. Findings were consistent with overall data trends: 11 had viral copies at lower detection limit (LDL) or $< 40$ copies/ml and 3 had a viral load between 21 and 1000 copies/ml. We performed further sensitivity analyses in which we performed multiple imputation for missing data with the assumption that data was missing at random. By this method, results remained similar: 13 of 16 individuals with missing Tanner staging were Tanner 1 in imputed data, and 12 of 16 individuals with missing viral loads were LDL or $< 40$ copies/mL after imputation.

### Baseline measures

We first compared differences in QUS and DXA measurements between CLHIV and uninfected controls (Table 3). DXA measures of WB BMD and BMC, whole body BMD, were lower in CLHIV than controls, similar to our previous reports [2, 11]. Calcaneal SOS and SI were significantly lower in CLHIV, while radial SOS did not differ between groups (Table 3).

We performed power calculations to determine the expected effect size given the sample size for both calcaneal and radial QUS. With a predetermined power of 80 percent and alpha level of 0.05, the expected correlation coefficient for calcaneal QUS (n = 170) was 0.213, and the expected coefficient for radial QUS (n = 205) was 0.194 (S1 Fig in S1 File).

In analyses pooled across CLHIV and controls, calcaneal QUS measures were strongly correlated with DXA measures (Fig 1(a) and 1(b)). Both calcaneal BUA and SI were correlated with whole body (r = 0.43 to r = 0.52, p < 0.01), lumbar spine (r = 0.38 to r = 0.43, p < 0.01), and 1/3 distal radius (r = 0.41 to r = 0.52, p < 0.01) BMD and BMC measures. Calcaneal SOS

**Table 1. Demographic characteristics of participants with insight calcaneal QUS measurements.**

| | (%) | Baseline (Visit 1) | | 12-month Follow-up (Visit 2) | | |
|---|---|---|---|---|---|---|
| | CLHIV (n = 80) | HIV- Control (n = 90) | P-value | CLHIV (n = 80) | HIV- Control (n = 90) | P-value |
| Age (years) | | | | | | |
| Mean (SD) | 7.14 (1.37) | 7.29 (1.55) | 0.503 | 8.13 (1.38) | 8.38 (1.53) | 0.26 |
| Median (IQR) | 7.03 (4.62, 9.45) | 7.27 (5.01, 9.92) | | 7.99 (5.71, 10.3) | 8.41 (5.60, 11.2) | |
| Sex, n (%) | | | | | | |
| Male | 40 (50.0%) | 51 (56.7%) | 0.474 | 40 (50.0%) | 51 (56.7%) | 0.474 |
| Female | 40 (50.0%) | 39 (43.3%) | | 40 (50.0%) | 39 (43.3%) | |
| Weight (kg) | | | | | | |
| Mean (SD) | 21.7 (4.43) | 23.7 (5.39) | 0.007 | 24.0 (5.08) | 26.9 (6.30) | 0.001 |
| Median (IQR) | 21.3 (16.2, 26.3) | 23.1 (17.7, 28.4) | | 23.8 (18.1, 29.5) | 26.0 (18.6, 33.2) | |
| Weight-for-age z-score | | | | | | |
| Mean (SD) | -0.579 (1.02) | -1.06 (1.10) | 0.004 | -0.610 (1.00) | 0.0268 (1.13) | <0.001 |
| Median (IQR) | -0.560 (-1.98, 0.86) | -0.230 (-1.63, 1.17) | | -6.40 (-2.02, 0.74) | -0.115 (-1.57, 1.34) | |
| Underweight, n (%) | | | | | | |
| No | 73 (91.3%) | 88 (97.8%) | 0.12 | 68 (85.0%) | 73 (81.1%) | 0.121 |
| Yes | 7 (8.8%) | 2 (2.2%) | | 6 (7.5%) | 1 (1.1%) | |
| Missing | | | | 6 (7.5%) | 16 (17.8%) | |
| Height (cm) | | | | | | |
| Mean (SD) | 115 (8.37) | 119 (8.50) | 0.008 | 121 (7.87) | 126 (7.94) | <0.001 |
| Median (IQR) | 115 (103, 128) | 120 (108, 132) | | 120 (109, 132) | 126 (115, 137) | |
| Height-for-age z-score | | | | | | |
| Mean (SD) | -1.21 (0.913) | -0.716 (0.792) | <0.001 | -1.06 (0.957) | -0.554 (0.822) | <0.001 |
| Median (IQR) | -1.36 (-2.67, -0.05) | -0.665 (-1.85, 0.52) | | -1.15 (-2.29, -0.002) | -0.465 (-1.63, 0.70) | |
| Stunted, n (%) | | | | | | |
| No | 62 (77.5%) | 84 (93.3%) | 0.006 | 70 (87.5%) | 86 (95.6%) | 0.104 |
| Yes | 18 (22.5%) | 6 (6.7%) | | 10 (12.5%) | 4 (4.4%) | |
| BMI | | | | | | |
| Mean (SD) | 16.1 (1.95) | 16.6 (2.31) | 0.135 | 16.1 (1.95) | 16.9 (2.71) | 0.046 |
| Median (IQR) | 15.8 (14.1, 17.5) | 16.2 (13.6, 18.7) | | 16.0 (13.6, 18.4) | 16.4 (13.4, 19.5) | |
| BMI-for-age z-score | | | | | | |
| Mean (SD) | 0.230 (1.07) | 0.429 (1.19) | 0.25 | 0.0489 (0.990) | 0.304 (1.30) | 0.149 |
| Median (IQR) | 0.245 (-0.76, 1.25) | 0.305 (-1.18, 1.97) | | 0.080 (-0.96, 1.12) | 0.335 (-1.41, 2.08) | |
| Tanner Stage, n (%) | | | | | | |
| Tanner 1 | 78 (97.5%) | 88 (97.8%) | 1 | 76 (95.0%) | 66 (73.3%) | 0.172 |
| Tanner 2 | 2 (2.5%) | 2 (2.2%) | | 3 (3.8%) | 8 (8.9%) | |
| Missing | | | | 1 (1.3%) | 16 (17.8%) | |
| Viral load (copies/ml) [N (%)] | | | | | | |
| LDL (≤20 or ≤40) | 73 (91.3%) | NA | NA | 53 (66.2%) | NA | NA |
| 21–1000 | 6 (7.5%) | | | 9 (11.2%) | | |
| >1000 | 1 (1.25%) | | | 2 (2.50%) | | |
| N.A. | 0 (0.0%) | | | 16 (20.0%) | | |
| CD4+ count (cells/µl), mean (SD) | 1185.5 (373) | NA | NA | 1122.7 (393) | NA | NA |
| ART regimen category [N (%)] | | | | | | |
| EFV-based | 41 (51.2%) | NA | NA | 43 (53.7%) | NA | NA |
| LPV/r-based | 39 (48.7%) | | | 37 (46.2%) | | |

**Table 2. Demographic characteristics of participants with Mini-Omnisense QUS measurements.**

| | Baseline (Visit 1) | | | 12-month Follow-up (Visit 2) | | |
|---|---|---|---|---|---|---|
| | CLHIV (n = 101) | HIV- Control (n = 104) | P-value | CLHIV (n = 101) | HIV- Control (n = 104) | P-value |
| Age (years) | | | | | | |
| Mean (SD) | 6.57 (1.32) | 6.94 (1.45) | 0.059 | 7.57 (1.33) | 7.98 (1.44) | 0.035 |
| Median (Min, Max) | 6.12 (5.01, 9.77) | 6.83 (5.02, 9.99) | | 7.12 (6.01, 11.1) | 7.86 (6.01, 11.0) | |
| Sex, n (%) | | | | | | |
| Male | 50 (49.5%) | 55 (52.9%) | 0.731 | 50 (49.5%) | 55 (52.9%) | 0.731 |
| Female | 51 (50.5%) | 49 (47.1%) | | 51 (50.5%) | 49 (47.1%) | |
| Weight (kg) | | | | | | |
| Mean (SD) | 19.5 (4.14) | 22.2 (5.22) | <0.001 | 21.8 (4.69) | 25.1 (6.32) | <0.001 |
| Median (Min, Max) | 18.8 (13.8, 39.9 | 21.4 (14.6, 48.0) | | 21.1 (14.8, 46.6) | 24.2 (16.7, 54.5) | |
| Weight-adjusted *z*-score | | | | | | |
| Mean (SD) | -0.900 (0.902) | -0.281 (0.974) | <0.001 | -0.858 (0.891) | -0.179 (1.03) | <0.001 |
| Median (Min, Max) | -0.970 (-2.46, 1.94) | -0.375 (-2.31, 3.64) | | -0.810 (-2.60, 1.57) | -0.270 (-2.57, 3.47) | |
| Underweight, n (%) | | | | | | |
| No | 91 (90.1%) | 103 (99.0%) | 0.011 | 86 (85.1%) | 91 (87.5%) | 0.026 |
| Yes | 10 (9.9%) | 1 (1.0%) | | 9 (8.9%) | 1 (1.0%) | |
| Missing | | | | 6 (5.9%) | 12 (11.5%) | |
| Height (cm) | | | | | | |
| Mean (SD) | 112 (8.93) | 117 (9.27) | <0.001 | 118 (8.27) | 124 (8.90) | <0.001 |
| Median (Min, Max) | 110 (94.6, 137) | 118 (95.7, 140) | | 117 (98.5, 142) | 124 (104, 147) | |
| Height-adjusted *z*-score | | | | | | |
| Mean (SD) | -1.36 (0.925) | -0.675 (0.920) | <0.001 | -1.21 (0.897) | -0.472 (0.954) | <0.001 |
| Median (Min, Max) | -1.45 (-3.31, 1.08) | -0.695 (-3.73, 2.11) | | -1.28 (-3.36, 1.35) | -0.610 (-3.17, 2.50) | |
| Stunted, n (%) | | | | | | |
| No | 76 (75.2%) | 96 (92.3%) | 0.002 | 83 (82.2%) | 101 (97.1%) | <0.001 |
| Yes | 25 (24.8%) | 8 (7.7%) | | 18 (17.8%) | 3 (2.9%) | |
| BMI-for-age *z*-score | | | | | | |
| Mean (SD) | -0.0675 | 0.163 (1.05) | 0.101 | -0.152 (0.922) | 0.0174 (1.13) | 0.241 |
| Median (Min, Max) | (0.946) | 0.155 (-2.10, 3.31) | | -0.160 (-2.61, 2.00) | -0.0300 (-3.20, 3.84) | |
| | -0.0800 (-3.37, 1.87) | | | | | |
| Tanner Stage, n (%) | | | | | | |
| Tanner 1 | 100 (99.0%) | 102 (98.1%) | 1 | 98 (97.0%) | 88 (84.6%) | 0.428 |
| Tanner 2 | 1 (1.0%) | 2 (1.9%) | | 3 (3.0%) | 6 (5.8%) | |
| Missing | | | | | 10 (9.6%) | |
| Viral load (copies/ml) [N (%)] | | | | | | |
| LDL (≤20 or ≤40) | 93 (92.1%) | NA | NA | 59 (58.4%) | NA | NA |
| 21–1000 | 5 (4.95%) | | | 12 (11.9%) | | |
| >1000 | 3 (2.97%) | | | 4 (3.96%) | | |
| N.A. | 0 (0.0%) | | | 26 (25.7%) | | |
| CD4+ count (cells/μl), mean (SD) | 1281.2 (459) | NA | NA | 1185.0 (386) | NA | NA |
| ART regimen category [N (%)] | | | | | | |
| EFV-based | 54 (53.4%) | NA | NA | 57 (56.4%) | NA | NA |
| LPV/r-based | 47 (46.5%) | | | 44 (43.6%) | | |

**Table 3. Comparison of baseline and follow-up QUS and DXA measures between CLHIV and uninfected controls.**

| | Calcaneal QUS | | | Radial QUS | | |
|---|---|---|---|---|---|---|
| | **CLHIV (n = 80)** | **HIV- Control (n = 90)** | **p-value** | **HIV+ (n = 101)** | **HIV- Control (n = 104)** | **p-value** |
| **Baseline** | | | | | | |
| SOS, mean (SD) | 1558.9 (17.6) | 1568.8 (19.7) | <0.001 | 3609.3 (117) | 3587.7 (128) | 0.213 |
| BUA, mean (SD) | 85.15 (11.8) | 88.75 (13.1) | 0.061 | NA | NA | NA |
| Stiffness Index, mean (SD) | 73.26 (10.0) | 78.57 (11.5) | 0.002 | NA | NA | NA |
| Whole Body BMC, mean (SD) | 715.0 (118) | 783.8 (133) | <0.001 | 671.3 (121) | 770.7 (141) | <0.001 |
| Whole Body BMD, mean (SD) | 0.673 (0.063) | 0.708 (0.064) | <0.001 | 0.652 (0.065) | 0.701 (0.067) | <0.001 |
| Lumbar BMC, mean (SD) | 17.31 (5.13) | 18.52 (5.16) | 0.126 | 15.63 (4.67) | 17.29 (5.02) | 0.015 |
| Lumbar BMD, mean (SD) | 0.600 (0.071) | 0.619 (0.066) | 0.078 | 0.586 (0.076) | 0.610 (0.068) | 0.016 |
| 1/3 Distal Radius BMC, mean (SD) | 36.99 (9.04) | 39.78 (9.53) | 0.052 | 32.95 (8.61) | 38.23 (10.0) | <0.001 |
| 1/3 Distal Radius BMD, mean (SD) | 0.406 (0.056) | 0.421 (0.061) | 0.099 | 0.380 (0.058) | 0.405 (0.063) | 0.003 |
| **Follow-up** | | | | | | |
| SOS, mean (SD) | 1567.1 (20.9) | 1571.4 (20.2) | 0.175 | 3623.9 (105) | 3570.6 (134) | 0.002 |
| BUA, mean (SD) | 88.78 (13.7) | 97.88 (13.6) | <0.001 | NA | NA | NA |
| Stiffness Index, mean (SD) | 78.09 (12.2) | 85.33 (11.0) | <0.001 | NA | NA | NA |
| Whole Body BMC, mean (SD) | 813.9 (131) | 877.2 (139) | 0.003 | 762.0 (133) | 858.7 (147) | <0.001 |
| Whole Body BMD, mean (SD) | 0.729 (0.064) | 0.754 (0.062) | 0.012 | 0.704 (0.064) | 0.743 (0.069) | <0.001 |
| Lumbar BMC, mean (SD) | 20.20 (5.29) | 21.26 (5.64) | 0.212 | 18.43 (4.94) | 20.08 (5.52) | 0.026 |
| Lumbar BMD, mean (SD) | 0.612 (0.074) | 0.632 (0.072) | 0.076 | 0.592 (0.074) | 0.626 (0.068) | <0.001 |
| 1/3 Distal Radius BMC, mean (SD) | 42.52 (9.30) | 46.24 (10.3) | 0.015 | 38.22 (8.89) | 44.10 (11.1) | <0.001 |
| 1/3 Distal Radius BMD, mean (SD) | 0.439 (0.060) | 0.453 (0.062) | 0.150 | 0.412 (0.057) | 0.436 (0.065) | 0.006 |

did not correlate with most DXA measures and displayed only modest correlations with whole body BMC (r = 0.15, p < 0.01), and lumbar spine BMC (r = 0.19, p < 0.01).

Similarly, SOS from the radial QUS correlated with whole body (r = 0.26 to r = 0.31, p < 0.01), lumbar spine (r = 0.19 to r = 0.28, p < 0.01), and 1/3 distal radius (r = 0.29 to r = 0.33, p < 0.01) BMD and BMC measures.

We additionally explored the baseline correlations between calcaneal QUS and DXA in analyses stratified by HIV status. Results of stratified correlations were comparable to the main pooled analysis, with BUA and SI showing strong correlations with BMD and BMC, particularly among CLHIV (S3 and S4 Figs in S1 File).

## Follow-up measures

We also determined differences between groups in terms of QUS and DXA measures at the 12-month follow-up visit, and results were largely similar. While significant at baseline, calcaneal SOS did not differ at 12-months. Calcaneal BUA and SI were significantly lower in CLHIV, while radial SOS was significantly higher in CLHIV, at follow-up.

We also performed cross-sectional correlations at 12-months, and calcaneal QUS was strongly correlated with DXA measures (Fig 1(c) and 1(d)). Calcaneal BUA and SI were each correlated with whole body, lumbar spine and 1/3 distal radius BMC and BMD (r = 0.25 to r = 0.45, p < 0.001). Additionally, BUA was correlated with lumbar and 1/3 distal radius (p < 0.05). Calcaneal SI was correlated with 1/3 distal radius BMC and BMD (p < 0.05). Calcaneal SOS displayed modest correlations with whole body BMC and BMD, as well as lumbar BMC (p < 0.05). At follow-up, 1/3 distal radius SOS was modestly correlated with 1/3 distal radius BMD, as well as whole body BMC and BMD (p < 0.05).

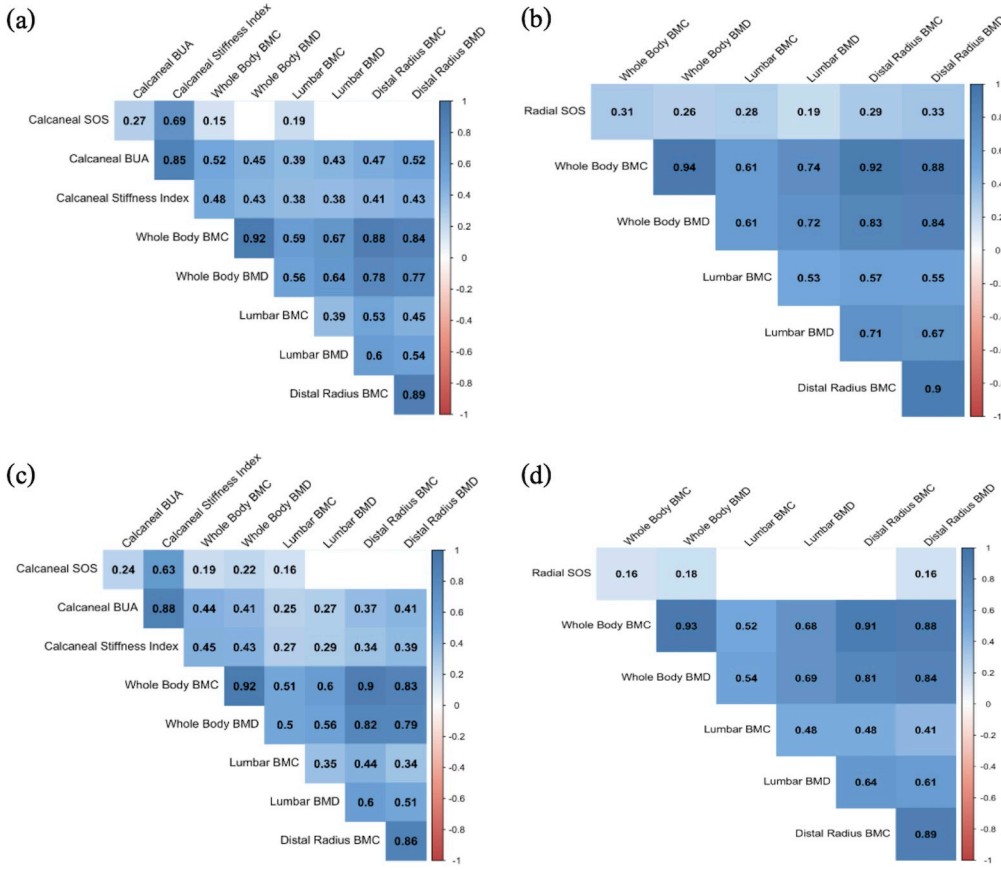

**Fig 1. Correlations between QUS and DXA measures at baseline and follow-up, pooled across HIV groups.** (a) Results of Spearman correlation between baseline calcaneal QUS measures and DXA (n = 170). (b) Results of Spearman correlation between baseline radial QUS measures and DXA (n = 205). Shaded cells indicate $p < 0.05$, with darker shading indicating more significant values. (c) Results of Spearman correlation between follow-up calcaneal QUS measures and DXA (n = 170). (d) Results of Spearman correlation between follow-up QUS measures and DXA (n = 205). Shaded cells indicate $p < 0.05$, with darker shading indicating more significant values. Numbers included within individual cells represent Spearman rho values. BMC: bone mineral content; BMD: bone mineral density; BUA: broadband ultrasound attenuation; SOS: speed of sound; DXA: dual-energy x-ray absorptiometry.

We additionally explored the follow-up correlations between calcaneal QUS and DXA in analyses stratified by HIV status. Results of stratified correlations were comparable to the main pooled analysis, with BUA and SI showing strong correlations with DXA, particularly whole-body BMD and BMC (S3 and S4 Figs in S1 File).

## Longitudinal measures

Between-group differences were also determined between the absolute change from baseline to follow-up in QUS and DXA measures. Change in whole body BMD differed between groups, with CLHIV demonstrating a greater increase in whole body BMD over the 12-month period. CLHIV also demonstrated a greater increase in calcaneal SOS during the study period. To determine whether changes in calcaneal and radial QUS measures correlated with changes in DXA measures between visits, we performed Spearman correlations between the absolute change between visits of each measurement (S2 Fig in S1 File).

Considering absolute change, calcaneal QUS measures were poorly correlated with DXA measures. Only change in calcaneal SOS had a modest correlation with change whole body

BMC and BMD (r = 0.17 to r = 0.22, p < 0.05). Change in radial SOS was also poorly correlated with change in BMC and BMD.

As in the main pooled analyses, correlations between absolute change in calcaneal QUS and DXA were not consistently identified in analyses stratified to CLHIV only or uninfected controls only (S3 and S4 Figs in S1 File).

## Inter-visit correlations

We also examined the correlation of calcaneal BUA and whole body DXA BMD between baseline and follow-up visits to determine the degree of within-subject variability for each variable. Considering the correlation between baseline and follow-up for whole body DXA BMD, strong positive correlations were identified for both CLHIV (rho = 0.924, p < 0.001) and control (rho = 0.921, p < 0.001) groups (Fig 2(a) and 2(b)). In contrast, weaker positive correlations were identified for calcaneal BUA between baseline and follow-up visits in CLHIV (rho = 0.318, p = 0.004) and control (rho = 0.508, p < 0.001) groups (Fig 2(c) and 2(d)).

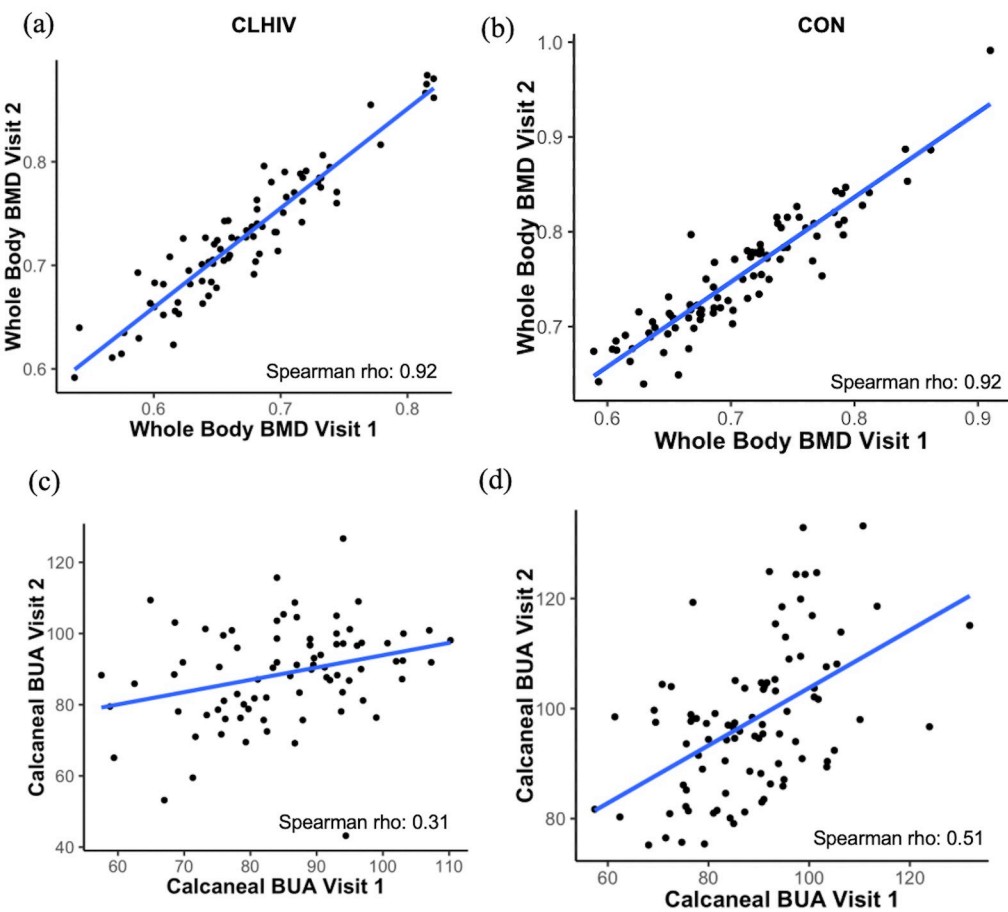

**Fig 2. Within-subject Spearman correlations between baseline (Visit 1) and follow-up (Visit 2) visits, stratified by study group.** (a) Correlation between baseline and follow-up whole body DXA BMD within CLHIV group. (b) Correlation between baseline and follow-up whole body DXA BMD within control group. (c) Correlation between baseline and follow-up calcaneal BUA. (d) Correlation between baseline and follow-up calcaneal BUA. Blue lines indicate the line of best fit from Spearman correlation. BMD: bone mineral density; BUA: broadband ultrasound attenuation; DXA: dual-energy x-ray absorptiometry.

## QUS comparisons across DXA z-score groups

To determine if differences in QUS measurements were present in those with abnormal BMD, we stratified the cohort into groups according to z-scores at each visit (S5 Table in S1 File). At baseline, those with a DXA whole-body BMD $z \leq$ -1.0 (n = 61) had significantly lower calcaneal BUA and SI compared to those with $z >$ -1.0 (n = 109), and these findings were recapitulated at 12-months. No difference was identified between calcaneal SOS between these groups. There were no differences in radial QUS parameters between those with whole body DXA BMD $z \leq$ -2.0 and $z >$ 2.0 at either baseline or follow-up. Similarly, there were no differences in QUS parameters between those who demonstrated a decrease in whole body DXA BMD z-score between baseline and 12-months at either visit.

## Discussion

In this longitudinal analysis of the relationships between calcaneal and radial QUS and DXS, we found a strong correlation between calcaneal QUS and DXA at two cross-sectional time points. In particular, both calcaneal BUA and SI were strongly correlated with whole body DXA measures at baseline and 12-month follow-up. However, neither calcaneal nor radial QUS displayed strong correlations with DXA when absolute change between baseline and follow-up was considered. In addition, while DXA BMD displayed strong inter-visit correlations within subjects, calcaneal BUA displayed much weaker, albeit significant, correlations between visits.

Several studies have explored the relationship between QUS and DXA measures in different study populations. Among post-menopausal women, for instance, a strong correlation has consistently been found between calcaneal QUS and DXA at multiple body sites [8, 26]. Similarly, a study of adult patients living with HIV identified more moderate correlations between calcaneal QUS and DXA, particularly whole body DXA measurements, which is consistent with the present study [7]. More variable results are reported in children. Similar to our study, calcaneal SI correlated with whole body BMC and BMD in a cohort of healthy Chinese children and adolescents [9]. Likewise, SOS measured by tibial QUS in a healthy Dutch pediatric cohort correlated strongly with both lumbar and whole body DXA, and a Swedish pediatric study of healthy participants identified strong correlations of calcaneal SI, SOS, and BUA with whole body and lumbar BMD [27, 28]. These studies provide strong evidence that the correspondence between calcaneal QUS and DXA, particularly whole body DXA, is significant and may have clinical utility.

More inconsistent results have been reported with respect to radial QUS, similar to the results we report. In a prior study of healthy Malaysian children, agreement between radial QUS and DXA was weak [29]. This finding was recapitulated in a study comparing radial QUS to DXA among healthy Thai children [30]. A study of previously pre-term infants also failed to identify a correlation between radial SOS and DXA measures [31]. Similarly, in children with hemophilia A, no correlations were identified between radial QUS measures and BMD [32]. As such, the poor agreement we observed between radial QUS and DXA, even at the same measurement site, suggests that the two methods are not comparable for bone status assessment in CLHIV. In our study, this discordance may further be explained by our measurement of only SOS at the radius, a parameter which was not well correlated with DXA at the calcaneus and likely reflects different properties of bone than does BUA.

Despite strong cross-sectional correlations, changes in BUA and SI failed to correlate with changes in DXA measures. This may be due to the fact that QUS outputs such as BUA measure different properties of bone than DXA, and these properties may display different trajectories of growth compared to BMC and BMD measured by DXA. BUA reflects not only the quantity

of bone through which the ultrasound beam passes but also the physical arrangement, or micro-architecture, of the bone [33, 34], which may also partially explain why BUA is a better predictor of fracture risk than DXA in some populations [7, 35]. As DXA is unable to distinguish between trabecular and cortical bone, changes in DXA measures may not sensitively reflect some of the micro-architectural changes that occur during periods of growth in childhood and adolescence [36]. Notably, calcaneal BUA displayed a much weaker inter-visit correlation than did DXA BMD, suggesting the measure may not be as stable as DXA over time. Some have suggested that while QUS may adequately measure response to therapy, and correlates well with DXA, it may lack precision in repeat measures longitudinally [37]. Others have described the need for more precise quality control measures for QUS, such as those existing for DXA, that would improve the precision of QUS as a longitudinal measure, particularly noting that QUS devices may be sensitive to temperature changes and require frequent calibration [38–40].

Taken together, our results build upon prior work to suggest the utility of calcaneal QUS for assessment of bone status in pre-pubertal South African CLHIV. We identify that in cross-section, BUA is a strong correlate of DXA, yet differences in which aspects of bone architecture each methodology captures may render longitudinal comparisons less reliable. Furthermore, the finding of poor inter-visit correlation in BUA compared to DXA suggests the ongoing need for improvements in precision and quality control in QUS methodology. Regardless, in RCS where DXA is unlikely to become widely available and in which the majority of CLHIV reside, calcaneal ultrasound may provide a useful tool for initial assessment of bone quality.

## Supporting information

**S1 File.**
(DOCX)

**S1 Dataset. Supporting dataset.**
(XLSX)

## Acknowledgments

We thank all the participants and staff at the Empilweni Services and Research Unit.

## Author Contributions

**Conceptualization:** Stephanie Shiau, Stephen M. Arpadi, Michael T. Yin.

**Data curation:** Yanhan Shen, Renate Strehlau, Stephen M. Arpadi.

**Formal analysis:** Jackson A. Roberts, Jonathan J. Kaufman.

**Funding acquisition:** Stephen M. Arpadi, Michael T. Yin.

**Investigation:** Jackson A. Roberts, Yanhan Shen, Renate Strehlau, Faeezah Patel, Louise Kuhn, Ashraf Coovadia, Jonathan J. Kaufman, Stephanie Shiau, Stephen M. Arpadi, Michael T. Yin.

**Methodology:** Renate Strehlau, Faeezah Patel, Stephanie Shiau, Stephen M. Arpadi, Michael T. Yin.

**Project administration:** Faeezah Patel, Louise Kuhn, Ashraf Coovadia.

**Resources:** Ashraf Coovadia, Stephen M. Arpadi, Michael T. Yin.

**Supervision:** Stephanie Shiau, Stephen M. Arpadi, Michael T. Yin.

**Visualization:** Jackson A. Roberts.

**Writing – original draft:** Jackson A. Roberts.

**Writing – review & editing:** Yanhan Shen, Renate Strehlau, Jonathan J. Kaufman, Stephanie Shiau, Stephen M. Arpadi, Michael T. Yin.

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
