## [Decision Letter · Decision Letter 0]

3 Aug 2022

PONE-D-22-15788Comparison of Quantitative Ultrasonography and Dual X-ray Absorptiometry for Bone Status Assessment in South African Children Living with HIVPLOS ONE

Dear Dr. Roberts,

Thank you for submitting your manuscript to PLOS ONE. After careful consideration, we feel that it has merit but does not fully meet PLOS ONE’s publication criteria as it currently stands. Therefore, we invite you to submit a revised version of the manuscript that addresses the points raised during the review process.

Both reviewers have requested additional detail regarding the analytical approach.  Please provide this.

We look forward to receiving your revised manuscript.

Kind regards,

Robert Daniel Blank, MD, PhD

Academic Editor

PLOS ONE

Journal Requirements:

Reviewers' comments:

Reviewer's Responses to Questions

**Comments to the Author**

1. Is the manuscript technically sound, and do the data support the conclusions?

Reviewer #1: Partly

Reviewer #2: Yes

2. Has the statistical analysis been performed appropriately and rigorously? 

Reviewer #1: No

Reviewer #2: Yes

3. Have the authors made all data underlying the findings in their manuscript fully available?

Reviewer #1: No

Reviewer #2: Yes

4. Is the manuscript presented in an intelligible fashion and written in standard English?

Reviewer #1: Yes

Reviewer #2: Yes

5. Review Comments to the Author

Reviewer #1: The manuscript conducts secondary statistical analysis to assess correlations, both cross-sectional and longitudinal, between two different methods of QUS and DXA on data generated from a previous noninferiority RCT comparing CLHIV versus controls without HIV. There are 2 longitudinal time-points: baseline, and 12-month followup. The study is based on prior science. I have some additional questions.

(a) I understand this is secondary analysis, but a sample size/power justification would assist readers to plan future trials and analysis like this. Authors may simply try to calculate the effect size they wanted to see (at the onset) to assess the differences between the groups.

(b) LOCF is really a sub-optimal method in considering missing data imputation; with several techniques available, the authors may resort to one of the credible ones. A team statistician (within the author list) is recommended, and who can be helpful in using the correct imputation method.

Reviewer #2: This reviewer has a methods concern:

1. The DXA scores were grouped: Z-score groups were z <-1.0, z < -1.5 and those who had declining z-score between visits

Please explain why the data were grouped by z-score less than -1.0 and less than -1.5. Why wasn’t the z-score of <-2.0 used, which is the internationally accepted level of what is considered to be low bone density in children. This reviewer is not convinced that there would be enough difference between the z-score less than -1.0 and less than -1.5 groups to be meaningful, particularly in this sample size.

Doubtful that this will impact the overall findings of the paper, but these groups, based on z-score, seem arbitrary and do not follow any guidelines for what is considered to be low BMD. They do not sensibly stratify the data.

2. Editorial comment: z-scores should be expressed as a decimal (e.g. -1.0). In this paper, z-scores are expressed as -1.

6. PLOS authors have the option to publish the peer review history of their article (what does this mean?). If published, this will include your full peer review and any attached files.

Reviewer #1: No

Reviewer #2: No

---

## [Author Response · Author response to Decision Letter 0]

10 Sep 2022

The authors would like to thank the reviewers for their insightful comments, which we feel have substantially strengthened the manuscript. We have incorporated all suggested edits and have included responses to each individual reviewer’s comments below.

Reviewer #1: The manuscript conducts secondary statistical analysis to assess correlations, both cross-sectional and longitudinal, between two different methods of QUS and DXA on data generated from a previous noninferiority RCT comparing CLHIV versus controls without HIV. There are 2 longitudinal time-points: baseline, and 12-month followup. The study is based on prior science. I have some additional questions.

(a) I understand this is secondary analysis, but a sample size/power justification would assist readers to plan future trials and analysis like this. Authors may simply try to calculate the effect size they wanted to see (at the onset) to assess the differences between the groups.

To address this concern, we have calculated the effect size using the pwr package in R, as described in Lines 205-207. Estimated effect coefficients are included in Lines 277-280. Supplementary Figure 1 has also been added and includes a visual output of power calculations with expected effect size given our sample size and power of 80%.

(b) LOCF is really a sub-optimal method in considering missing data imputation; with several techniques available, the authors may resort to one of the credible ones. A team statistician (within the author list) is recommended, and who can be helpful in using the correct imputation method.

To address this, we have conducted an additional sensitivity analysis in which we performed multiple imputation assuming data was missing at random and have added the details to the Methods section (Lines 200-202). Results of this second sensitivity analysis were consistent with those from the last observation carried forward method, as well as the main analysis (Lines 261-265).

Reviewer #2: This reviewer has a methods concern:

1. The DXA scores were grouped: Z-score groups were z <-1.0, z < -1.5 and those who had declining z-score between visits

Please explain why the data were grouped by z-score less than -1.0 and less than -1.5. Why wasn’t the z-score of <-2.0 used, which is the internationally accepted level of what is considered to be low bone density in children. This reviewer is not convinced that there would be enough difference between the z-score less than -1.0 and less than -1.5 groups to be meaningful, particularly in this sample size.

Doubtful that this will impact the overall findings of the paper, but these groups, based on z-score, seem arbitrary and do not follow any guidelines for what is considered to be low BMD. They do not sensibly stratify the data.

We have amended the methods and results to utilize cutoffs of z <= -2, in accordance with the latest consensus guidelines, and have removed the z <= -1.5 cutoff. As there are a very small number of participants in this study with z <= -2, we have utilized the z <= -1 cutoff to capture a larger number of participants with reduced DXA scores for comparison that did not necessarily meet consensus guidelines for childhood osteoporosis. We have reflected the Methods (Lines 218-219) and Results (Lines 379-380) to reflect these changes. 

2. Editorial comment: z-scores should be expressed as a decimal (e.g. -1.0). In this paper, z-scores are expressed as -1.

We have edited the paper to include decimals for z-scores.

---

## [Decision Letter · Decision Letter 1]

5 Oct 2022

Comparison of Quantitative Ultrasonography and Dual X-ray Absorptiometry for Bone Status Assessment in South African Children Living with HIV

PONE-D-22-15788R1

Dear Dr. Roberts,

We’re pleased to inform you that your manuscript has been judged scientifically suitable for publication and will be formally accepted for publication once it meets all outstanding technical requirements.

Kind regards,

Robert Daniel Blank, MD, PhD

Academic Editor

PLOS ONE

Additional Editor Comments (optional):

Reviewers' comments:

Reviewer's Responses to Questions

**Comments to the Author**

1. If the authors have adequately addressed your comments raised in a previous round of review and you feel that this manuscript is now acceptable for publication, you may indicate that here to bypass the “Comments to the Author” section, enter your conflict of interest statement in the “Confidential to Editor” section, and submit your "Accept" recommendation.

Reviewer #1: All comments have been addressed

Reviewer #2: All comments have been addressed

2. Is the manuscript technically sound, and do the data support the conclusions?

Reviewer #1: (No Response)

Reviewer #2: Yes

3. Has the statistical analysis been performed appropriately and rigorously? 

Reviewer #1: (No Response)

Reviewer #2: Yes

4. Have the authors made all data underlying the findings in their manuscript fully available?

Reviewer #1: (No Response)

Reviewer #2: Yes

5. Is the manuscript presented in an intelligible fashion and written in standard English?

Reviewer #1: (No Response)

Reviewer #2: Yes

6. Review Comments to the Author

Reviewer #1: (No Response)

Reviewer #2: (No Response)

7. PLOS authors have the option to publish the peer review history of their article (what does this mean?). If published, this will include your full peer review and any attached files.

Reviewer #1: No

Reviewer #2: No

---

## [Editor Report · Acceptance letter]

7 Oct 2022

PONE-D-22-15788R1 

Comparison of Quantitative Ultrasonography and Dual X-ray Absorptiometry for Bone Status Assessment in South African Children Living with HIV 

Dear Dr. Roberts:

I'm pleased to inform you that your manuscript has been deemed suitable for publication in PLOS ONE. Congratulations! Your manuscript is now with our production department. 

Kind regards, 

on behalf of

Professor Robert Daniel Blank 

Academic Editor

PLOS ONE